

# The apparent permeabilities of Caco-2 cells to marketed drugs: magnitude, and independence from both biophysical properties and endogenite similarities

Steve O'Hagan and Douglas B. Kell

School of Chemistry & The Manchester Institute of Biotechnology and Centre for Synthetic Biology of Fine and Speciality Chemicals (SYNBIOCHEM), The University of Manchester, Manchester, Lancs, United Kingdom

## ABSTRACT

We bring together fifteen, nonredundant, tabulated collections (amounting to 696 separate measurements) of the apparent permeability ($P_{app}$) of Caco-2 cells to marketed drugs. While in some cases there are some significant interlaboratory disparities, most are quite minor. Most drugs are not especially permeable through Caco-2 cells, with the median $P_{app}$ value being some $16 \cdot 10^{-6}$ cm s$^{-1}$. This value is considerably lower than those (1,310 and $230 \cdot 10^{-6}$ cm s$^{-1}$) recently used in some simulations that purported to show that $P_{app}$ values were too great to be transporter-mediated only. While these values are outliers, all values, and especially the comparatively low values normally observed, are entirely consistent with transporter-only mediated uptake, with no need to invoke phospholipid bilayer diffusion. The apparent permeability of Caco-2 cells to marketed drugs is poorly correlated with either simple biophysical properties, the extent of molecular similarity to endogenous metabolites (endogenites), or any specific substructural properties. In particular, the octanol:water partition coefficient, log$P$, shows negligible correlation with Caco-2 permeability. The data are best explained on the basis that most drugs enter (and exit) Caco-2 cells via a multiplicity of transporters of comparatively weak specificity.

## INTRODUCTION

Most pharmaceutical drugs, and all oral ones, must necessarily cross at least one cell membrane to act. Understanding how this transport is effected remains a major challenge (*Kell & Oliver, 2014*). We have brought together considerable published evidence (e.g., *Dobson & Kell, 2008*; *Kell, 2013*; *Kell, 2015*; *Kell et al., 2013*; *Kell, Dobson & Oliver, 2011*; *Kell & Oliver, 2014*) that suggests that (in contrast to the general textbook belief, e.g., *Avdeef, 2012*; *Cao, Yu & Sun, 2006*; *Krogsgaard-Larsen, Liljefors & Madsen, 1996*; *Van De Waterbeemd & Testa, 2009*) small molecule drugs 'hitchhike' on the many protein transporters (*Kell, 2013*; *Kell & Goodacre, 2014*; *Sahoo et al., 2014*; *Thiele et al., 2013*) that are part of normal intermediary

Corresponding author
Douglas B. Kell,
dbk@manchester.ac.uk

metabolism. These transporters may be identified via experiments where gene expression levels are manipulated systematically as independent variables (*César-Razquin et al., 2015*; *Giacomini et al., 2010*; *Han et al., 2015*; *Kell & Oliver, 2014*; *Lanthaler et al., 2011*; *Winter et al., 2014*). A number of recent books summarise the importance of protein transport to drug disposition (*Bhardwaj et al., 2008*; *Ecker & Chiba, 2009*; *Fromm & Kim, 2011*; *Ishikawa, Kim & König, 2013*; *Sugiyama & Steffansen, 2013*; *You & Morris, 2014*).

Caco-2 cells (e.g., *Artursson, Palm & Luthman, 2001*; *Awortwe, Fasinu & Rosenkranz, 2014*; *Balimane & Chong, 2005*; *Fearn & Hirst, 2006*; *Feng et al., 2014*; *Hidalgo, Raub & Borchardt, 1989*; *Sarmento et al., 2012*; *Sun et al., 2008*; *Van Breemen & Li, 2005*; *Volpe, 2011*) are an epithelial cell line that has become a *de facto* standard in studies of pharmaceutical drug transport. They form a more or less (and otherwise) impermeable layer that is polarised, in the sense of having 'apical' and 'basolateral' faces in which transporters are differentially expressed. They express hundreds of transporters (*Anderle, Huang & Sadée, 2004*; *Hayeshi et al., 2008*; *Landowski et al., 2004*; *Pshezhetsky et al., 2007*; *Sun et al., 2002*), and (although far from perfect (*Hilgendorf et al., 2007*)) they have significant predictive power as to the fraction of oral dose absorbed in humans (e.g., *Marino et al., 2005*; *Rubas et al., 1996*).

It is thus of general interest to understand the kinds of apparent permeability ($P_{app}$) rates for different drug molecules that Caco-2 cells can sustain. Although there are undoubtedly larger databases in-house in commercial and other enterprises, we have sought to bring together what we can of published data to determine the kinds of permeability values that Caco-2 cells can sustain, and what might determine that. We recognise that many factors can affect a specific measurement, e.g., the seeding density, age of the cells, pH and so on. An interlaboratory comparison (*Hayeshi et al., 2008*) indicated that while on occasion measurements could vary by more than an order of magnitude, overall the groupings were normally reasonably tight (say within a factor of 2–5).

The question of $P_{app}$ values in Caco-2 cells has been brought into sharper focus by a recent article (*Matsson et al., 2015a*; *Matsson et al., in press*) that claimed unusually high rates for verapamil and propranolol, based on measurements in a specific earlier article (*Avdeef et al., 2005*) in which stirring had been performed at a massive rate (and one not used in any equivalent transporter kinetics measurements). We indicated that these values were major outliers (by one or even two orders of magnitude) (*Mendes, Oliver & Kell, 2015*), but did not pursue the question of what might be typical values of $P_{app}$ for other drugs. This is the focus of what we do here.

## METHODS

Data were extracted manually from tables in the papers stated, and compiled as an Excel sheet. Typical biophysical descriptors were added using the RDKit module (*Riniker & Landrum, 2013*) of KNIME (*Berthold et al., 2008*; *Mazanetz et al., 2012*; *Saubern, Guha & Baell, 2011*) (www.knime.org/), essentially as described (*O'Hagan & Kell, 2015a*; *O'Hagan & Kell, 2015b*; *O'Hagan et al., 2015*). For one experiment we used the CDK-KNIME nodes (*Beisken et al., 2013*).

We have selected a set of 15 studies (indicated in the legend to Fig. 1) for our analysis. Based on the list of FDA-approved drugs that we downloaded (as before (*O'Hagan & Kell, 2015b*; *O'Hagan et al., 2015*)) from DrugBank (http://drugbank.ca) (*Law et al., 2014*), we compiled from these a non-redundant set of measurements of the apparent permeability ($P_{app}$, that are commonly given in units of cm s$^{-1}$). Although there are older papers, we have started with the compilation of Hou and colleagues (*2004*). Our method for avoiding redundancy in later compilations was not to include a separate measurement if the numbers given were identical to those in *Hou et al. (2004)* (or any other later papers) to at least 1 decimal place. We ignore any efflux transporters, since the evidence (that we show later) is that their influence on these measurements is fairly small (*Lin et al., 2011*). We incorporated two values from the review of Marino and colleagues (*2005*), one from lower throughput 24-well plates, one from a 96-well assay.

Where data were available for bidirectional assays, e.g., *Hayeshi et al. (2008)* and *Skolnik et al. (2010)*, they are given just for the $A \rightarrow B$ direction. In the case of the interlaboratory comparison (*Hayeshi et al., 2008*), we used solely 'batch 1' data, while in the work of *Lin et al. (2011)*, efflux inhibitors were sometimes present, as noted below. The entire dataset is given as an Excel sheet as a Table S1, and consists of 696 separate measurements. As indicated in Methods, we used KNIME to append some simple biophysical descriptors.

## RESULTS

Figure 1A shows all of the data, with those studies finding rates above $100 \cdot 10^{-6}$ cm s$^{-1}$ labelled with the study number. Of the 21 measurements that have this property, no fewer than 9 (labelled in red) are from a study (*Avdeef et al., 2005*) of Avdeef and colleagues. The largest values (*Avdeef et al., 2005*) were observed at very high values of stirring rates (700 rpm), and these in particular contained a great many outliers. The implication is that these increases at exceptionally high stirring rates were due to unstirred layer effects, although it is hard to see their relevance to *in vivo* drug absorption where no such stirring is occurring. We also note (*Dahlgren et al., 2015*; *Fagerholm & Lennernäs, 1995*) that stirring has no effect on the transport of drugs through actual intestines. Mannitol is sometimes used as a membrane-impermeant control, taken to pass via a paracellular route. This said, mannitol controls did not always have the lowest values, and inulin (*Marino et al., 2005*) or EDTA (*Lin et al., 2011*) may be better. Although it was stated (*Avdeef et al., 2005*) that mannitol transport rates were 'normal', it is unclear why they do not change with stirring rates (or whether they do), so it is not entirely certain whether the epithelial layer remained intact, especially at some of the highest stirring rates employed. For these and other reasons, and especially given the strongly outlying nature of the measurements, we have decided for the rest of the analysis to exclude the data from *Avdeef et al. (2005)*, resulting in an overall dataset of 680 separate measurements as shown in Fig. 1B. Although the $P_{app}$ values might vary somewhat with the drug concentrations (e.g., *Engman et al., 2003*), we made no systematic attempt to take this into account, since (i) often the drug concentration values appearing in the Tables from which we took the data were not actually given, and (ii) this would not be expected to be by more than a factor 2, well within the

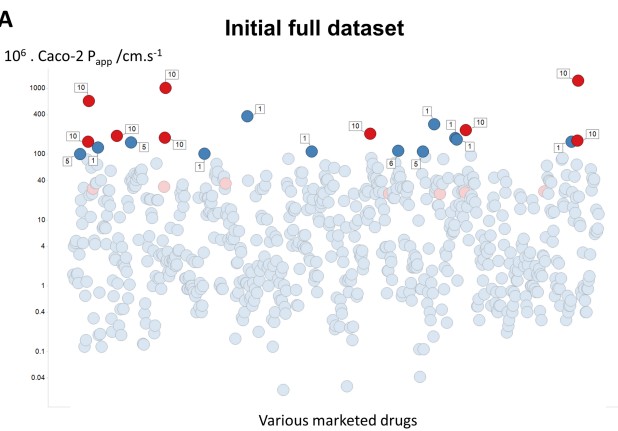

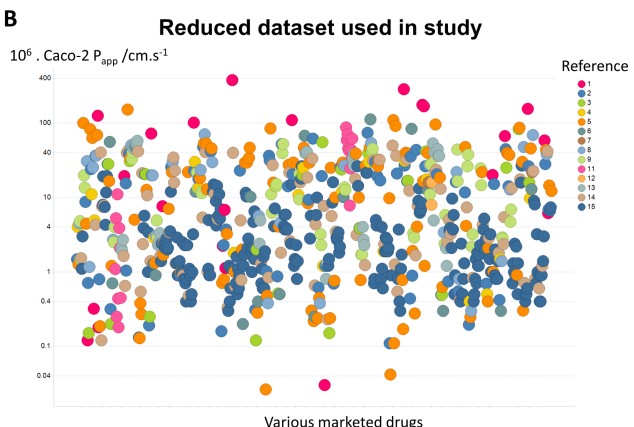

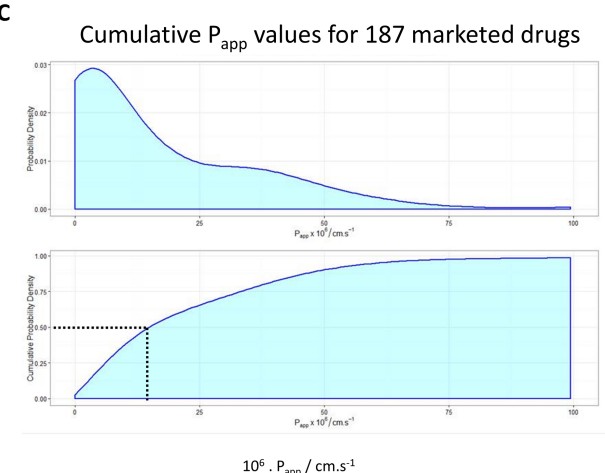

**Figure 1 A compilation of 15 review articles on Caco-2 permeability measurements.** (A) Full dataset, including outliers. (B) Reduced dataset after removal of the data from *Avdeef et al. (2005)*. (C) Cumulative plot and smoothed histogram of the Caco-2 permeabilities in the reduced dataset. In (C) data for identical drugs were averaged. Data were extracted from the 
## Effects of efflux inhibitors on Caco-2 permeability

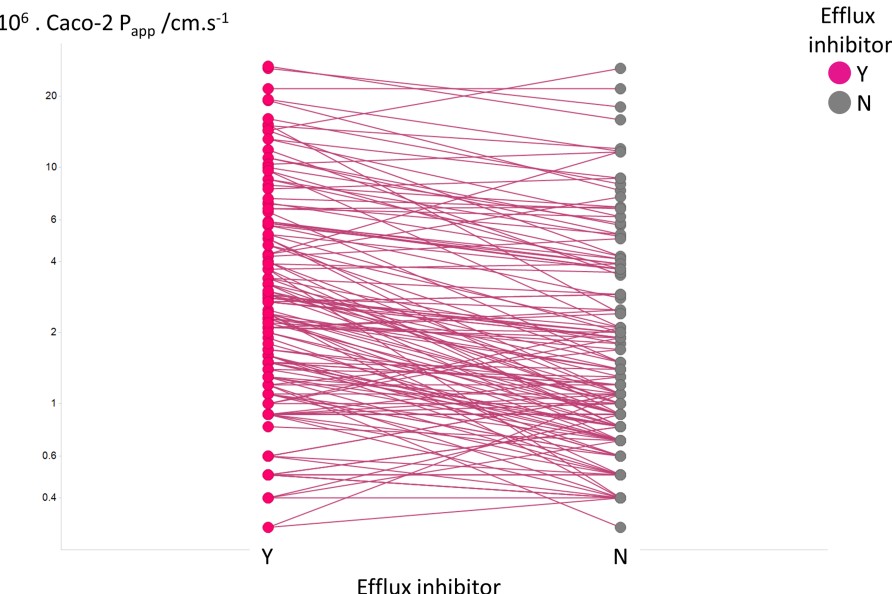

**Figure 2 Relative lack of effect of efflux inhibitors on Caco-2 permeabilities of marketed drugs.** Data are taken from *Lin et al. (2011)* and shown as paired values.

range of variation seen in individual measurements. A cumulative plot and smoothed histogram of the data (Fig. 1C) shows that the most abundant values for $P_{app}$ are in the range $3–4 \cdot 10^{-6}$ cm s$^{-1}$, and with a median value of ca $16 \cdot 10^{-6}$ cm s$^{-1}$. Obviously these values are considerably lower than those discussed in *Matsson et al. (2015a)* and *Matsson et al. (in press)*, and indicate (*Mendes, Oliver & Kell, 2015*) that typical transporter kinetic parameters and expression levels are entirely adequate to account alone for cellular drug uptake, as proposed (*Dobson et al., 2009*; *Dobson & Kell, 2008*; *Kell, 2013*; *Kell, 2015*; *Kell & Dobson, 2009*; *Kell et al., 2013*; *Kell, Dobson & Oliver, 2011*; *Kell & Goodacre, 2014*; *Kell & Oliver, 2014*; *Kell et al., 2015*).

The chief point of this high-level, overview paper is that the values of $P_{app}$ observed are typically rather low relative to those that can easily be explained on the basis of transporter-mediation only, without delving into minutiae. However, at the request of a reviewer we have added a Table (Table 1) that shows where available the concentrations of drug, insert type and stirring rates used in the relevant paper.

Figure 2 illustrates another feature of the data. Here we took the tabulated data of Lin and colleagues (*2011*) that used a variety of efflux inhibitors. A comparison showed that no very substantial (order-of-magnitude) differences in uptake were observed (Fig. 2),

**Table 1 Further details of the 15 transporter studies reviewed.**

| Drug concentration(s) | Insert type | Shaking or stirring speeds given | Reference |
|---|---|---|---|
| 0.02–6 mM | Polycarbonate filter inserts, 12 mm diameter; pore size 0.4 µm; Costar | Mainly 500 rpm | *Bergström et al. (2003)* |
| Compilation of 13 references; not possible to deconstruct | | | *Hou et al. (2004)* |
| Not actually stated | Polycarbonate filters (area 1.13 cm$^2$) in Costar Snapwell six-well plates | Not stated | *Corti et al. (2006)* |
| 100–200 µM | Corning 24-well polycarbonate filter membrane (HTS-Transwell inserts, surface area: 0.33 cm$^2$) | Not stated | *Balimane, Han & Chong (2006)* |
| 10 µM | Fibrillar collagen coated PET membrane inserts in 24-well plates (BD Biosciences) | Not stated | *Gozalbes et al. (2011)* |
| 10 µM | Collagen-coated 24-transwell plates | 100 rpm | *Peng et al. (2014)* |
| 1–10 µM | 24-well systems from BD BioSciences (PET membrane, 1.0 mm, cat. #351181) or Costar (polycarbonate membrane, 0.4 mm, cat. #3396). | 30 rpm | *Press (2011)* |
| Compilation; not possible to deconstruct | | | *Usansky & Sinko (2005)* |
| 5 µM | 'Filter membranes' | Not stated | *Marino et al. (2005)* |
| 50 nM–100 µM | Polycarbonate filter inserts (Transwell® Costar; mean pore size 0.45 µm; diameters 12 mm) | 25–7,000 rpm | *Avdeef et al. (2005)* |
| Mostly 30 µM, occasionally 100 µM | Polycarbonate, 0.4–3 µm, 6 mm or 12 mm | Not stated | *Hayeshi et al. (2008)* |
| 10–500 µM | 12-well Transwell plate with clear polyester membrane insert (0.4 µm pore diameter, 12 mm diameter), Corning Costar | 50 rpm | *Wang et al. (2010)* |
| 20 µM | 'Collagen-coated inserts' | Not stated | *Uchida et al. (2009)* |
| 10 µM | "96-Multiwell Insert System from BD Biosciences" | Yes, but rate not stated | *Skolnik et al. (2010)* |
| 10 µM | Six-well Transwell polycarbonate membrane inserts, Corning Life Science | Not stated | *Lin et al. (2011)* |

such that the typical 'low' values of $P_{app}$ cannot realistically be ascribed to a major role of efflux pumps.

## Lack of relationship between Caco-2 permeability values and simple biophysical properties of drugs

If unstirred layer effects and pure diffusion (as opposed to transporter-based enzyme kinetics) were significant in Caco-2 permeability (notwithstanding the evidence that they are not (*Fagerholm & Lennernäs, 1995*)), one might suppose that permeability values should depend significantly upon the molecular mass of the drug involved. However, Fig. 3A shows that this is not the case, as the line of best fit has a slope of only −0.04X and a value for $r^2$ of just 0.069. In a similar vein, despite a widespread view that transport rates should depend on log$P$, Fig. 3B shows that even when the Caco-2 permeabilities are plotted in log space, the $r^2$ value for a plot against $SlogP$ is only 0.011. (For a plot in linear space the value drops to just $r^2 = 0.004$, data not shown.) There is a slightly clearer relationship between Caco-2 permeability and a drug's total polar surface area, but again the relationship is fairly weak ($r^2 = 0.334$ when the ordinate is in log space, Fig. 3C,

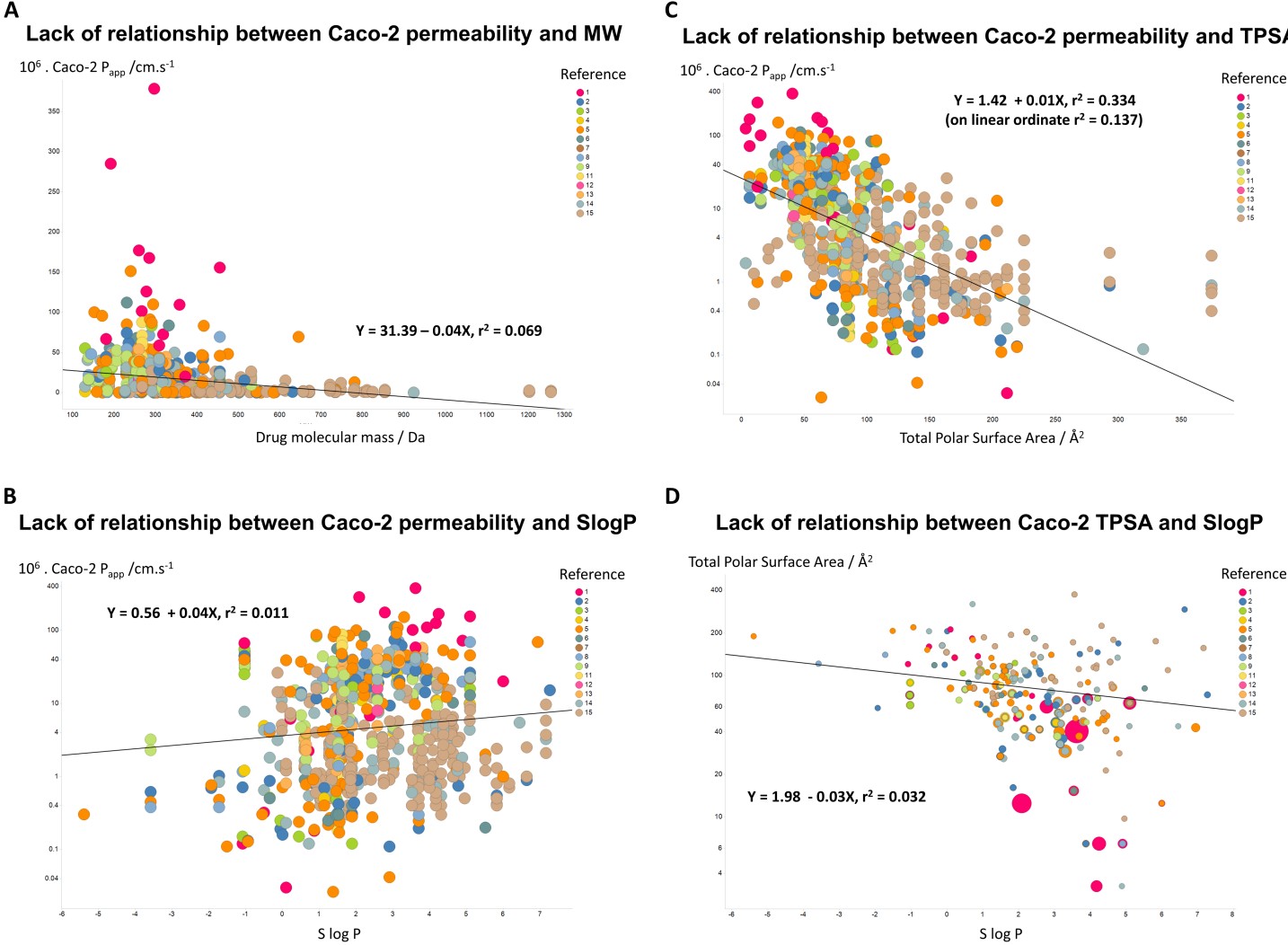

**Figure 3** **Lack of relationship between Caco-2 cells and simple biophysical parameters.** (A) Caco-2 permeability as a function of MW. (B) Caco-2 permeability as a function of $S\log P$. (C) Caco-2 permeability as a function of Total Polar Surface Area. (D) Lack of relationship between Total Polar Surface Area and $S\log P$. 5

but only $r^2 = 0.137$ when the ordinate is in linear space (plot not shown)). It is also of interest that there is no significant relationship between total Polar Surface Area and $S\log P$ (Fig. 3D). In particular, as before, we (e.g., *Dobson & Kell, 2008*; *Kell & Oliver, 2014*) and others (e.g., *Skolnik et al., 2010*) find that transmembrane permeability cannot be accounted for in terms of simple biophysical properties, and certainly not via $\log P$.

## Lack of relationship between Caco-2 permeability and structural similarity to endogenous metabolites

Since the natural role of the transporters that drugs hitchhike on is to transport endogenous metabolites (*Dobson & Kell, 2008*; *Kell, 2013*; *Kell, 2015*; *Kell et al., 2013*; *Kell & Oliver, 2014*; *Nigam, 2015*; *Swainston, Mendes & Kell, 2013*), the 'principle of molecular similarity' (e.g., *Bender & Glen, 2004*; *Eckert & Bajorath, 2007*; *Gasteiger, 2003*; *Maldonado et al., 2006*)

suggests that drugs should bear structural similarities to endogenous metabolites, and this is found to be the case (*Dobson, Patel & Kell, 2009*; *O'Hagan & Kell, 2015b*; *O'Hagan et al., 2015*). This led us to wonder whether any aspects of 'metabolite-likeness' might be related to Caco-2 permeability. However, we found no simple relationship of this type, whether (as illustrated) in terms of the closest Tanimoto similarity (Fig. 4A) or (for the 61 molecules for which this was true) the count of endogenites exceeding a Tanimoto similarity of 0.65 (Fig. 4B). (There was a very weak positive correlation, $r^2 = 0.156$, with the number of endogenites exceeding a Tanimoto similarity of 0.75, for the 21 molecules that had at least one, data not shown.) One interpretation of this is that while in some cases a rather small number of transporters are typically involved in drug uptake (e.g., *Winter et al., 2014*), in many cases a considerably greater number contribute (e.g., *Kell et al., 2013*; *Lanthaler et al., 2011*). While well enough known in general (*Mestres & Gregori-Puigjané, 2009*), such 'promiscuity' has become much more manifest using modern chemical biology approaches to detect protein binding directly (e.g., *Li et al., 2010*; *Niphakis et al., 2015*).

Finally, we wondered whether a standard machine learning approach (a random forest learner (*Breiman, 2001*; *Fernández-Delgado et al., 2014*; *Knight et al., 2009*; *O'Hagan & Kell, 2015b*)) might be able to predict Caco-2 permeabilities using a couple of fingerprint methods for encoding drug structures. Even this very powerful method had negligible predictive power as judged by its out-of-bag error (Fig. 5). It must be concluded that the ability to pass through Caco-2 cells is a very heterogeneous property, that cannot be accounted for via simple biophysical properties (e.g., those contributing to log $P$), and is best explained by the intermediacy of a very heterogeneous set of transporters.

## DISCUSSION AND CONCLUSIONS

A recent publication (*Matsson et al., 2015a*; *Matsson et al., in press*), using exceptionally high values of $P_{app}$ for verapamil and propranolol, claimed that the apparent permeability values were such that they could not be supported by known (random) transporters at random expression level, $K_m$ and $k_{cat}$ values. It was stated (*Matsson et al., 2015a*) that such rates "are possible in the absence of transmembrane diffusion, but only under very specific conditions that rarely or never occur for known human drug transporters". While we showed that this was simply not the case (quite the opposite) (*Mendes, Oliver & Kell, 2015*), it prompted us to ask the question as to what typical rates of $P_{app}$ might be for marketed drugs in Caco-2 cells more generally. By bringing together tabulated data from 15 studies, we found that the commonest values are just ca 3–4 · $10^{-6}$ cm s$^{-1}$, and that the median value is ca 16 · $10^{-6}$ cm s$^{-1}$. Thus, transporters alone can easily account for these. There was no significant correlation of $P_{app}$ values with either the values of various biophysical descriptors or measures of endogenite-likeness, and even powerful machine learning methods could not predict the permeabilities from the drug structures. The most obvious reason for this is simply that there is no unitary explanation (such as simplistic phospholipid bilayer diffusion), as most drugs exploit multiple but often unknown transporters with overlapping specificities. Which they are and how much each contributes to a given Caco-2 permeability must be determined by varying their activities

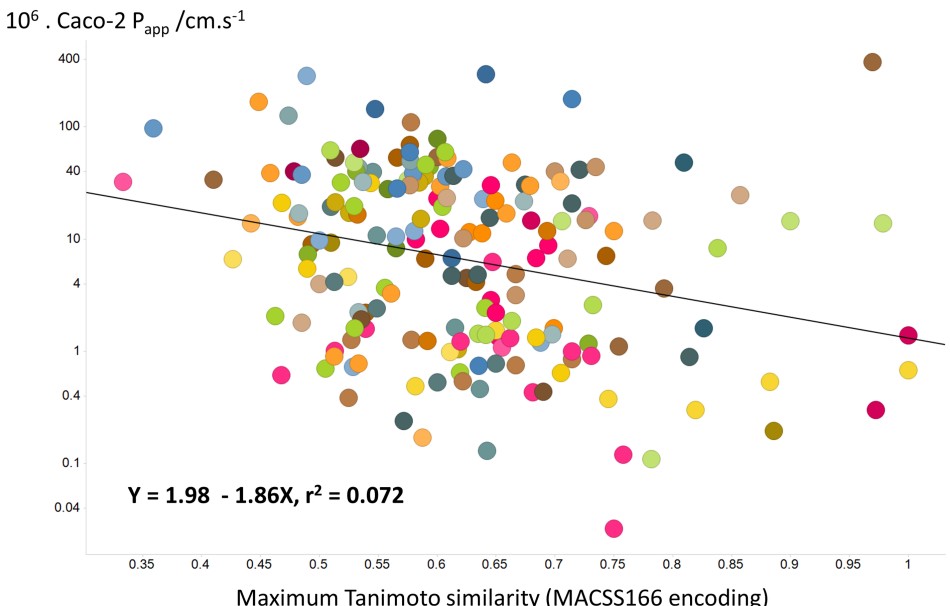

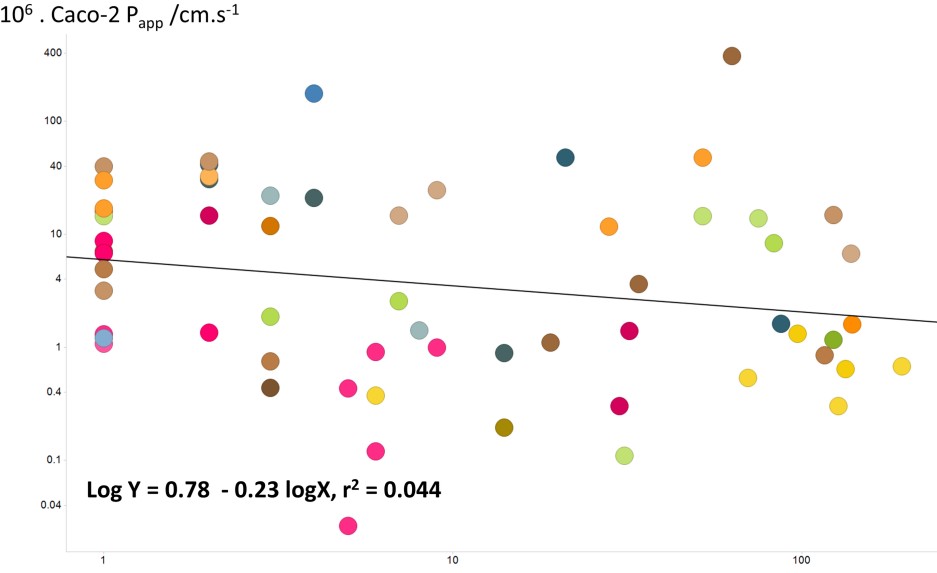

**Figure 4 Lack of relationship between Caco-2 cell permeability and measures of endogenite-likeness.** (A) Lack of relationship between the $P_{app}$ of a drug in Caco-2 cells and its greatest Tanimoto similarity to any endogenite molecule in Recon2. (B) Lack of relationship between the $P_{app}$ of a drug and the number of endogenous metabolites (endogenites) in Recon2 possessing a Tanimoto similarity greater than 0.65. 187 different drugs were assessed in each case.

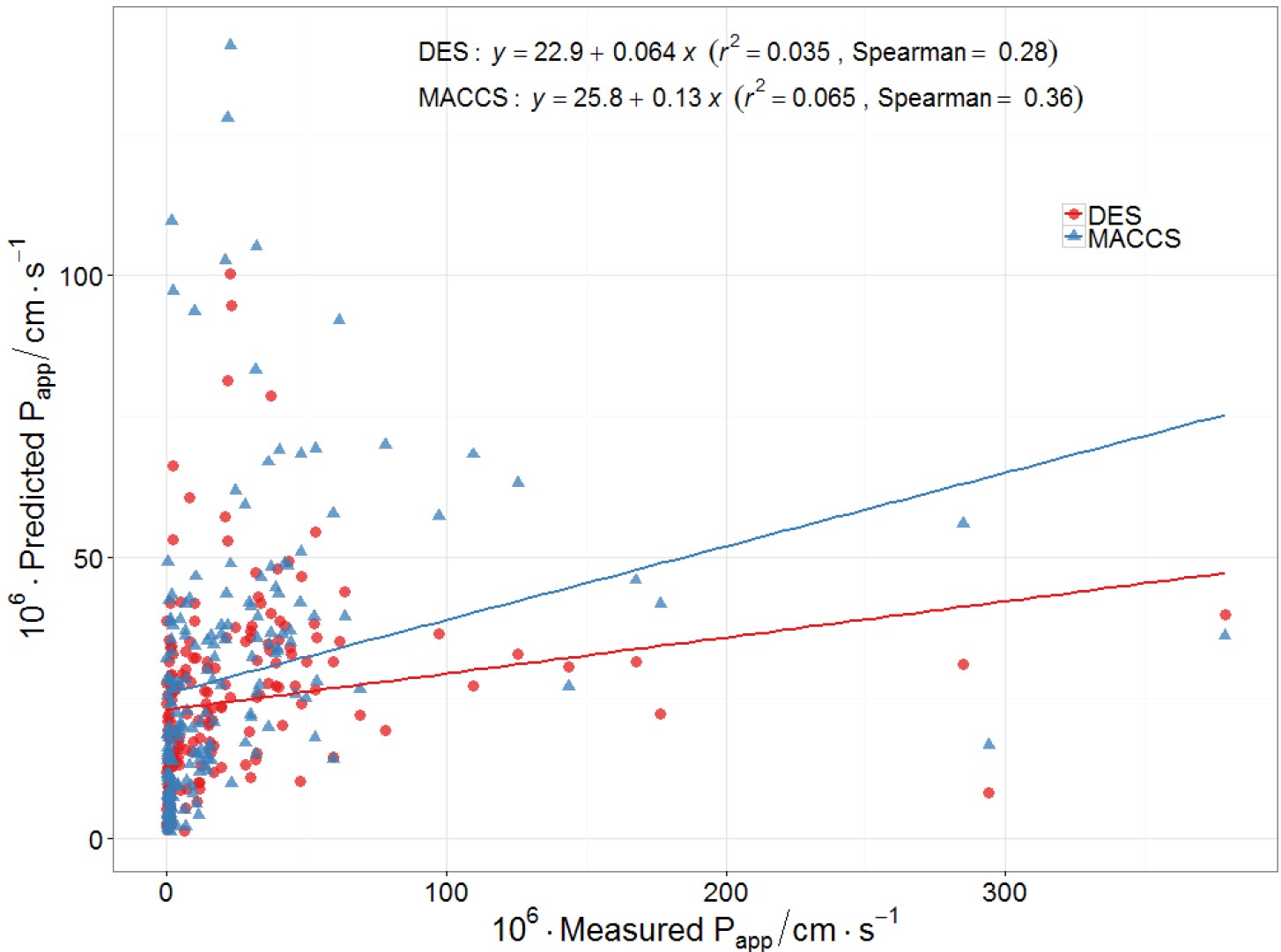

**Figure 5 Lack of relationship between experimental Caco-2 permeabilities and those predicted (via out-of-bag estimation) from a random forest learner.** Drug properties were encoded using either the MACCS166 encoding (*O'Hagan et al., 2015*) or the full DES encoding (*O'Hagan & Kell, 2015b*), each together with the molecular properties encoded in the CDK KNIME node (*Beisken et al., 2013*).

as independent variables (*Kell, 2015*; *Kell & Oliver, 2014*; *Kell et al., 2015*; *César-Razquin et al., 2015*), whether by using inhibitors (e.g., *Han et al., 2015*; *Ming et al., 2009*) or genetically. This latter activity has been initiated in other cell lines (e.g., *Giacomini et al., 2010*; *Han et al., 2015*; *Lanthaler et al., 2011*; *Winter et al., 2014*). The availability of powerful mammalian genome editing tools such as variants of the CRISPR/Cas9 system

(e.g., *Kleinstiver et al., 2015*; *Maeder et al., 2013*; *Wang et al., 2014*; *Zhou et al., 2014*) imply that we may soon expect to see this strategy applied with great effect to the Caco-2 system.

### Funding

The Biotechnology and Biological Sciences Research Council (BBSRC) provided financial support under grant BB/M017702/1. This is a contribution from the Centre for Synthetic Biology of Fine and Speciality Chemicals (SYNBIOCHEM). The funders had no role in study design, data collection and analysis, decision to publish, or preparation of the manuscript.

### Grant Disclosures

The following grant information was disclosed by the authors:
The Biotechnology and Biological Sciences Research Council: BB/M017702/1.
Centre for Synthetic Biology of Fine and Speciality Chemicals (SYNBIOCHEM).

### Competing Interests

The authors declare there are no competing interests.

### Author Contributions

- Steve O'Hagan and Douglas B. Kell conceived and designed the experiments, performed the experiments, analyzed the data, contributed reagents/materials/analysis tools, wrote the paper, prepared figures and/or tables, reviewed drafts of the paper.

### Data Availability

All the data we generated are provided in Supplemental Information 1.

### Supplemental Information

Supplemental information for this article can be found online at http://dx.doi.org/10.7717/peerj.1405#supplemental-information.

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
