# Peer review of "The apparent permeabilities of Caco-2 cells to marketed drugs: magnitude, and independence from both biophysical properties and endogenite similarities"

_PeerJ, doi:10.7717/peerj.1405_

## Round 0.1 · original submission · Major Revisions

The work is interesting and potentially relevant for the field, but the manuscript needs major amendments.

- introduction need to be rewritten
- parts of results need to be moved to methods
- concentration data for efflux transporter ligands
- PAMPA and Caco-2 model comparisons for available datasets

Reviewer 1 ·

Basic reporting

I find this manuscript interesting and worth to be published

Experimental design

No expeiments have been performed, but literature data have been screened and evuluated

Validity of the findings

The data appear to be robust, but sometimes they could have been a bit more detailed.
e.g. in figure 4 only a limited number of compounds has been studied. it would be interesting to see, which coumpounds show a polarized transport. Therefore, I suggest to set the names of the compounds at the right side of the image at the respective place.

Additional comments

see "validity of findings."

Reviewer 2 ·

Basic reporting

This paper described, in a non-innovative way, the apparent permeability of Caco-2 cells to marketed drugs, collected from 15 different research studies. The text is exaggeratedly supported by references from the authors’ previous works, substantially decreasing its interest. A notorious lack of concern regarding the report writing becomes evident though the use of a subjective (e.g. “more or less”) and non-scientific (e.g. “and so on”) speech. Despite the relevant prior literature, the introduction was written in a poorly convincing way, demonstrating fragilities to the reader (lines 58-61).

Experimental design

I have no comments regarding the experimental design. However, there is a mismanagement of the contents distribution throughout the ‘Methods’ and ‘Results’. Lines 80-96 must be included in the ‘Methods’ section.

Validity of the findings

The topic of the paper is very interesting, and the effort in bringing together results reported from different studies, is remarkable. It allowed broader interpretations of single results. However, the two main conclusions should be further supported, once they are based only in interpretations of results obtained from others, and the obtained findings promise to have major impacts on the field. These are the reasons why I would not recommend the acceptance of the paper yet.

Additional comments

No Comments.

Reviewer 3 ·

Basic reporting

The submitted manuscript contains interesting data on the apparent permeabilities of Caco-2 cells to marketed drugs and their different properties. The results are based on the collection of experimental data from 15 studies and databases and the authors correlated them to find relationship between permeabilities and biophysical or structural properties. The conception is excellent, however some important remarks are made up below.

Experimental design

The research question is not clearly defined. The introduction does not give enough background to understand the aim(s) of the work and how some results are connected to the introduction.
“Results” section contains descriptions of methodology until line 96, thus this part should be relocated to “Methods”.

Validity of the findings

An important experimental data is not considered during the evaluation of the results: the concentration of the drugs are not collected from the references. Concentration of the tested drugs is an important parameter in these experiments, especially in the case of efflux substrates. Some molecules (cyclosporine A, verapamil) are transported by P-glycoprotein at low concentrations, but they inhibit the function of this transporter at high concentrations. On the other hand Papp values can be calculated if the flux of the drug through the Caco-2 cell layer is constant in the function of time. High drug concentrations can destroy the cellular barrier and modify the drug flux, thus the concentration of tested drugs should be compared.
It is difficult to interpret Fig. 2. All of the drugs are put together in a figure, but there are molecules, which are substrates of efflux transporters and molecules which are not. Significant increases in Papp values can be observed in some cases in the presence of inhibitors, thus it would be better to arrange efflux substrates and non-substrates separately.
Conclusions are not appropriately stated. Unknown transporters are mentioned as the background of drug transport, but a very simple method to exclude the role of transporters is not investigated. PAMPA model is a simple model to exclude the role of transporters. It would be necessary to correlate the permeability values of PAMPA and Caco-2 models and see the role of simple phospholipid bilayer diffusion also.

Additional comments

No Comments.

---

## Round 0.2 · Minor Revisions

As the re-reviews were conflicting and not detailed enough to make a fair decision, two additional reviewers were invited. Their opinions were convincing about the values of the manuscript, but the reviews raised some important issues. Technical details (drug concentration, insert type and characteristics, assay time, stirring or not etc.) are very important for permeability studies and should be supplied as an additional table.

Reviewer 2 ·

Basic reporting

In general, no alterations were made. The introduction was not rewritten, still presenting low quality either in its content or scientific speech. The addition of references is not enough to turn the introduction acceptable.
Conclusions were not rethought as well.
No efforts were made on this new version to justify the acceptance of the paper.

Experimental design

The contents within "Methods" and "Results" sections were redistributed but no efforts were made to make the text clearer or more fluid.
Specifically, lack of concern becomes evident by stating "as indicated in Methods" (line 94) when the paragraph is now part of the Methods.

Validity of the findings

No alterations were made following reviewer's or editor's comments.

Additional comments

No comments.

Reviewer 3 ·

Basic reporting

I accept changes and answers.

Experimental design

No comments.

Validity of the findings

No comments.

Additional comments

No comments.

·

Basic reporting

No Comment

Experimental design

The method section seems to have been consolidated compare to the previous submission (at least what is in the method section of the revised manuscript is mainly an insertion) however the I would have appreciated a much more detailed description of the experimental data.

- When available in the original paper from which they were extracted the following information would be interresting to have without having to search in the original paper :
- Which drug concentration was used ? (when these information are available they should be mentionned)
- What was the duration of the drug transport experiment ?
- What kind of permeable inserts was used (size, pore size, membrane,etc..) ? (This is clearly missing as the mehod mentionned « We incorporated two values from the review of Marino and colleagues (Marino et al. 2005), one from lower throughput 24-well plates, one from a 96-well assay. » then I wonder what was the size of the assay for all other values ?
- What were the stirring conditions for the data other than Adveef et al. ? This is important to know as well as some discussion are based on the fact that « normal » stirring conditions have no effect on Papp.
- Some details on the Papp calculation would be also welcome

These informations could be added (as tables e.g as supplementary materials) like in Kell et al. 2015 in Trends in Pharmacological Sciences.

Validity of the findings

I think nobody doubt that some compounds are transported (influx or efflux) in Caco-2 whereas many people (including myself) have to be convinced that ‘phospholipid bilayer diffusion (of all drugs) is negligible.

The authors aims to reinforce this « concept » by demonstrating that no correlation could be found between biophysical properties or endogenite similarities and Caco-2 permeability.
This is clearly a risky business as strong a correlation (e.g between one physical characteristic and a biological value) is sometimes meaningfull whereas absence of correlation rarely prove that the this physical characteristic does not play a role in the observed biological data.

Indeed, I am curious to know if some values are predicted by some of the studied parameters (e.g strong correlation between Log P and Caco-2 Papp for let say 30 values among the dataset) in that case would the authors considered that this descriptor could be used for these compounds (i.e drugs) and not for the others ?

I am also wondering whether Papp values are the right values to adress the question of the dependence between Caco-2 permeability and biophysical properties. Indeed Papp does not account for the drug ability to permeate the inserts without cells (some other calculation does). Therefore, the validity of the different correlation would be based on the assomption that all drugs have the same permeability across the cell culture inserts (i.e without cells).
By experience, I have seen marked difference in permeability over the same inserts without cells for different compounds and even obtained lower permeability for the same compound in assay without cells (i.e empty inserts) with different format (e.g between 6 well and 12, 24 or 96 wells).

The argument that no systematic attempt to take the effect of drug concentration into account as « (ii) this would not be expected to be by more than a factor 2, well within the range of variation seen in individual measurements » sounds strange.
Indeed the authors promote that to permeate over Caco-2 « most drugs exploit multiple but often unknown transporters with overlapping specificities. »
Then wouldn’t the concentration used be crutial as some transporters might be saturated at some concentrations whereas other with low affinity for some compounds act only for high concentrations and not for lower concentrations ??

Additional comments

As I was not part of the previous round of reviewing I mainly evaluated the revised paper without paying too much attention to the previous round of reviewing.
Nevertheless I read the authors rebuttal letter and understood that the authors are sometimes having hard times to promote their ‘transporters-are-dominant’ view.

This situation is quite usual for people swimming against the tide and to therefore I recommend the publication of this paper as reviewing this manuscript gave me food for thought and could have the same effect on others.

Nevertheless, I have some concerns regarding the data presented and their interpretation which I encourage the authors to also consider.
I am sure that the authors are well aware of the fact that resisting prevailing opinion usually requires much more effort than following it.

I have no concern with revealing my identity to the authors and might come up with new thinking regarding their data later on but I would like them not to wait too long for my feed back.
Therefore I would appreciate if the authors could contact me if the paper got accepted and they decide to publish the full review history of their paper alongside the published article.

Reviewer 5 ·

Basic reporting

The article is well written and is a response to a secondary publication.

Experimental design

There is very little new data in this manuscript but the data they present is a compleling argument

Validity of the findings

THe paper is a theoretical description and should be interpreted with some caution

Additional comments

Overall I agree with the paper and understand the context of the rationale behind the manuscript. I would be cautious in making statements that lipoidal diffusion is negligible. Almost all drugs/molecules have some degree of passive permeation across the membrane.
The statements could be tempered a little.

---

## Round 0.3 · accepted · Accept

The comments of the second round of reviews were adequately answered.